# Plant-Derived Type I Ribosome Inactivating Protein-Based Targeted Toxins: A Review of the Clinical Experience

**DOI:** 10.3390/toxins14080563

**Published:** 2022-08-18

**Authors:** David J. Flavell, Sopsamorn U. Flavell

**Affiliations:** 1The Simon Flavell Leukaemia Research Laboratory, Southampton General Hospital, Southampton SO16 6YD, UK; 2Faculty of Medicine, University of Southampton, Southampton General Hospital, Southampton SO16 6YD, UK

**Keywords:** targeted toxins, ribosome inactivating proteins, clinical trials

## Abstract

Targeted toxins (TT) for cancer treatment are a class of hybrid biologic comprised of a targeting domain coupled chemically or genetically to a proteinaceous toxin payload. The targeting domain of the TT recognises and binds to a defined target molecule on the cancer cell surface, thereby delivering the toxin that is then required to internalise to an appropriate intracellular compartment in order to kill the target cancer cell. Toxins from several different sources have been investigated over the years, and the two TTs that have so far been licensed for clinical use in humans; both utilise bacterial toxins. Relatively few clinical studies have, however, been undertaken with TTs that utilise single-chain type I ribosome inactivating proteins (RIPs). This paper reviews the clinical experience that has so far been obtained for a range of TTs based on five different type I RIPs and concludes that the majority studied in early phase trials show significant clinical activity that justifies further clinical investigation. A range of practical issues relating to the further clinical development of TT’s are also covered briefly together with some suggested solutions to outstanding problems.

## 1. Introduction

Targeted toxins (TT) are, by definition, hybrid macromolecules that, through ligand recognition via an engineered binding domain, deliver a protein toxin payload to a target cell bearing the cognate ligand. The targeting domain of TTs can take several forms that include, but are not exclusive to, antibody-derived fragments, growth factors, cytokines and peptide sequences. First- and second-generation TTs were constructed using a variety of chemical conjugation methods. Subsequent later generation TTs are now more commonly constructed through the creation of chimeric fusion proteins using recombinant technology [1]. Early generation TTs suffered from inconsistent structural issues that led to the generation of a heterogeneous mixture of TT molecules [2]. This was a side effect of the semi-random nature of the basic chemical conjugation processes employed at the time. The presence of a heterogeneous mixture of molecular species in the final product made accurate toxicological and clinical evaluations of these therapeutic entities difficult. This molecular heterogeneity problem encountered in generation I and II TTs was largely overcome through the use of recombinant technology [3,4], and this aspect of clinical TT development will be expanded on in a later section.

Historically, the majority of TTs that have been investigated clinically have been based on ricin from seeds of the castor plant (*Ricinus communis*) [5] or on bacterial toxins derived from diphtheria [6] or pseudomonas [7]. Despite intensive studies spanning four decades, only two TTs have been approved and licenced for clinical use to date. Both are based on bacterial toxins: truncated pseudomonas toxin (PE38) or diphtheria toxin, respectively. Ontak (Denileukin Diftitox) is a fusion protein comprised of IL-2 and diphtheria toxin that targets the IL-2 receptor (CD25) and is approved for use in cutaneous T-cell lymphomas [8]. The recombinant immunotoxin (IT) BL22 (Moxetumomab pasudotox/Lumoxiti) is directed against the CD22 molecule expressed by a subpopulation of B cells and is indicated for use in cases of hairy cell leukaemia (HCL) [9]. The disappointing approval rate for TTs is despite the fact that several TTs studied clinically have demonstrated potent and selective cytotoxic activity, particularly in the case of haematological tumours. From our view, there are multiple reasons for this apparent translational failure that range from unacceptable drug toxicities (sometimes due to quality inconsistencies of the drug product) [10], immunogenicity issues [11] and perceived limitations to bulk drug production and formulation together with the often-transient nature of clinical responses observed in clinical trials with TTs when used as a monotherapy [12]. These various factors have all contributed to a general lack of commercial interest in TTs in favour of antibody drug conjugates (ADCs) [13,14] that are generally seen by the pharmaceutical industry as easier to manufacture and formulate, being based on small molecule cytotoxic payloads. These perceived issues have consequently impeded the effective clinical development of TTs, a fact that, from our view, is probably based on a false premise.

On the plus side, there are some major advantages that TTs have over the majority of ADCs that may in the future tip the balance in their favour. Firstly, the toxin payloads of TTs exert their cytotoxic action enzymatically, resulting in an increased potency (because ribosome inactivating proteins (RIPs) act enzymatically, relatively fewer toxin molecules need to gain access to the cytosol of the target cell to exert a potent cytotoxic effect). This is in comparison to small molecule cytotoxic payloads carried by ADCs that act stoichiometrically and therefore require a greater number of cytotoxic payload molecules to enter the target cell in order to exert their effect. Secondly, the fully proteinaceous nature of TTs lends itself to recombinant technology for design and at-scale production by fermentation, whereas ADCs, being chemical constructs, cannot exploit this route.

## 2. Clinical Studies with RIP-Based Targeted Toxins

This review deals only with the reported clinical experiences obtained for TTs constructed with plant-derived type I single-chain RIPs. The reader is referred to the review of Wayne et al. that covers a variety of other TTs that are based on several alternative toxins [15].

There are a vast number of plant-derived type I RIPs [16], but only five have been formally studied clinically when incorporated into various TTs. These are Saporin, from Soapwort (*Saponaria officianalis*) [17], Pokeweed Antiviral protein (PAP) from pokeweed (*Phytolacca Americana*) [18], Gelonin from Himalayan false lime (*Gelonium multiflorum*) [19], Momordin [20] from the balsam apple or pear (*Momordica balsamina)* and Bouganin from Bougainvillea or paper flower (*Bouggainvillea spectavilis Willd*) [21]. All have similar molecular weights (ca 30kD), and all lack a distinct ligand-binding domain. In their native forms, gelonin and PAP are both glycosylated, whereas saporin is not. All five RIPs possess N-glycosidase activity (EC 3.2.2.22) [22], whose action is to remove a single adenine residue from a GAGA sequence in the highly conserved α-sarcin loop contained within the 28S ribosomal RNA subunit [23,24,25]. This has the effect of irreversibly inhibiting protein synthesis by preventing an association between elongation factors and the ribosomal machinery. This event leads to apoptosis of the intoxicated cell via induction of the ribotoxic stress response [26] and other mechanisms that potentially include DNA damage [27], the unfolded protein response and interference with other signalling cascades [28].

As this review will reveal, there have been relatively few clinical studies undertaken with TTs constructed with single-chain RIPs, but those that have been described have, in the main results, displayed activity against the neoplasms for which they were designed. The purpose of this review is therefore to present the formal clinical studies that have been undertaken with TTs constructed with type I RIPs as summarised in Table 1. This, we hope, will give an overarching view of their potential value and provide food for thought to those decision-makers of the future who may well wish to make the next step with suitable resources and take this class of biologic drug forward in clinical development.

## 3. Saporin-Based TTs

The RIP saporin is derived from the soapwort plant (*Saponaria officianalis*) and is richly present in the seeds and leaves in a variety of different isoforms, the SO6 isoform from seeds being the most abundant and potent [41]. Only relatively few clinical studies with saporin-based TTs have been conducted and published, and the salient features of these studies together with those of three unpublished studies from our own group are briefly described here. All the described studies have used the SO6 isoform of saporin and mostly demonstrate acceptable toxicity profiles with some positive indications of TT activity against the diseases for which each was designed.

### 3.1. BER-H2-SO6 Immunotoxin (Anti-CD30)

The very first published clinical study with a saporin-based IT was described for patients with advanced therapy refractory Hodgkin’s lymphoma patients by Falini and co-workers in 1992 [29]. These workers employed an anti-CD30 whole IgG1 BER-H2 murine antibody chemically conjugated to the SO6 isoform of saporin, yielding an IT named BER-H2-SO6 [42]. BER-H2-SO6 IT, as described in previous in vitro and in vivo preclinical studies, shows selective and potent cytotoxicity for CD30^+^ cell lines [42,43], making this a good candidate therapeutic molecule for the treatment of Hodgkin’s disease (HD) and CD30^+^ large cell lymphomas [44,45]. Initially, a small series of four Hodgkin’s lymphoma patients with bulky advanced refractory disease were described following treatment with one single (0.8 mg/kg body weight) or two fractionated doses of 0.6 and 0.2 mg/kg given i.v. on days 1 and 7. In three of the patients, rapid and substantial reductions in all tumour masses ranging from 50 to 100% was observed, which persisted for 6–10 weeks; following which, tumour regrowth occurred at the original sites, with the exception of skin lesions on one patient. One patient treated with the fractionated doses on days 1 and 7 showed complete regression and was alive 10 months posttreatment. BER-H2-SO6 had significant antitumour activity with minimal systemic toxicity at the dose levels employed, the major issues being transient hepatotoxicity and thrombocytopenia in one patient with no instances of overt vascular leak syndrome (VLS). Human anti-mouse antibodies (HAMA) and human anti-saporin antibodies (HASA) were reported to occur in all four patients. This initial group of patients was then expanded to a total of 12, and the outline results of this study were reported briefly in a review paper and as a meeting abstract [46,47]. In the expanded phase I study, HD patients were treated with BER-H2-SO6 at dose levels ranging from 0.2 to 0.8 mg/m^2^ given as a single bolus injection. Rapid reductions in tumour masses ranging from 50% to >75% were reported in 40% of treated patients [47]. The average duration of the response was reported as 8 weeks, with side effects seen in 70% of treated patients that included fever, myalgias and VLS and four- to five-fold transient increases in the liver enzymes ALT and AST. The MTD was established as 0.8 mg/m^2^ based on a grade III VLS episode characterised by a pericardial effusion and pulmonary oedema. Despite these very promising phase I results reported for BER-H2-SO6, there are no further published reports of therapeutic clinical studies with this IT.

### 3.2. Bispecific Antibodies Targeting CD19 and CD22 in Chronic Lymphocytic and Lymphoma Patients

An alternative to the immunotoxin approach is to use a bispecific antibody (BsAb) to deliver saporin to a target cell population. In this instance, one Fab arm of the BsAb is directed against an epitope on the saporin molecule and the other Fab arm against an antigen on the target cell surface. The BsAb attaches univalently to the cell surface and forms a bridge to saporin that is internalised by receptor-mediated endocytosis (RME). In general, RME proceeds more efficiently when adjacent equivalent target molecules are co-ligated on the cell surface. Since BsAbs only attach to a target molecule univalently, internalisation is less efficient and can be significantly influenced by the characteristics of a particular target molecule, with some internalising better than others [30]. The authors of the clinical studies cited below sought to overcome this issue by using a combination of two different BsAbs that allowed for the cross-linking of two adjacent CD22 molecules via a saporin bridge.

The first description of a study to use a BsAb for saporin delivery in a clinical context was reported by Bonardi and co-workers in 1992 [48]. In this instance, BsAbs constructed with murine antibodies containing one Fab arm against either CD19 or CD22 and the other against saporin were investigated in just two patients with either end stage B-cell chronic lymphocytic leukaemia (CLL) or B-cell follicle centre centrocytic/centroblastic lymphoma.

In the case of the one CLL patient, BsAb RFB-9 × Sap1 was administered as four separate bolus injections given together with escalating doses (1, 2, 3 and 4 mg) of the SO6 isoform of saporin on days 0, 7, 28 and 42, totalling 100 mg BsAb and 10 mg of SO6 saporin over a 42-day period. No therapeutic effect was observed in this patient, and neither a HAMA or HASA response was detected over the duration of the study. There were no marked toxic side effects other than a mild headache, flu-like symptoms and a transient rise in temperature post-BsAb/saporin administration.

The second patient with B-cell follicle centre lymphoma was treated with three semi-escalating doses of SO6 saporin (1, 2 and 2 mg on days 0, 7 and 14) administered together with equal amounts of two different BsAbs (HD6 × anti-Sap5 and 4KB128 × Sap1) to a total BsAb dose level of 50 mg. Each BsAb possessed one Fab arm against human CD22 and the other Fab arm against saporin with the two different BsAbs, recognising distinctly different epitopes on the extracellular domain of CD22 and saporin. This combination of two different BsAbs allows for cooperative binding on the target cell surface via adjacent CD22 molecules and a resultant saporin bridge, as described by French et al. [49]. This mixture was administered as a 2-h i.v. infusion in 100 mL saline. No toxicities were observed other than some minor irritation in the administering vein during the infusion. There was a marked ten-fold reduction in circulating peripheral lymphocytes over the 27-day period of the study, with a complete resolution of ascites by the third treatment, together with reduction in the size of presumptively involved inguinal lymph nodes. By day 27, the patient developed a HAMA response that precluded any further treatment with BsAb.

The same group later reported a clinical study in two separate publications that described the same patients: four low-grade B-cell lymphomas [31] and the same four patients plus one additional B-CLL patient [49]. This study used a combination of a pair of BsAbs, each directed against CD22 in one Fab arm and saporin in the other but with each recognising a different nonoverlapping epitope on the CD22 and saporin molecules. This therefore theoretically allowed for the cross-linking of adjacent CD22 molecules on the cell surface via a saporin bridge, as described in previous preclinical work of his group. This effect gives rise to the greater binding affinity to both CD22 and saporin, with potential improvements in target cell internalisation of the BsAb-Saporin complex (see French et al. [49] for full details). In this clinical study, BsAbs and the SO6 isoform of saporin were premixed in a 3:1 molar ratio 24 h before administration to patients as a 1-h i.v. infusion at dose levels between 2 mg to 4 mg SO6 given weekly for between 3 and 6 weeks, except for a fifth patient who received the first three doses on a daily basis. The authors reported that the toxicities were minimal, with grade 2 myalgia and malaise being the most common. Local reversible erythematous inflammation along the vein used for administration of the BsAb/SO6 mixture was reported for two patients and occurred within 6 h of administration. Minor rises in the serum alanine aminotransferase (ALT) and creatinine levels were observed in two patients. A HAMA and HATA response was seen in only one patient at 14 and 21 days post-infusion, respectively. All five patients were reported to exhibit rapid and significant responses to the extent that at least a 50% reduction in a measurable disease was apparent in the week following their last treatment. However, all the therapeutic effects that included the elimination of circulating tumour cells, lymph node size reductions and improved bone marrow status, together with the elimination of pleural and ascitic effusions in two patients, were transient, since no response was sustained at this level for more than 28 days following the cessation of treatment, after which tumour progression occurred. Although different doses of saporin/BsAb were administered to a small group of patients, this study cannot be classified as a true dose escalation trial, and the maximum tolerated dose (MTD) was never established. It is therefore fair to say that the dose regime used was not optimal, and better longer-lasting therapeutic effects may be achievable with a dose schedule regime based on a true MTD. Nevertheless, this limited study showed that this approach gave demonstrably substantial selective cytotoxic effects against the target antigen-bearing tumour cells, leading the authors to conclude that this warranted further clinical investigation.

### 3.3. Phase I Study of the Anti-CD19 IT BU12-SAPORIN in B-Cell Follicular Lymphoma

BU12-SAPORIN is an IT comprised of the intact murine anti-human CD19 IgG_1_ antibody, BU12, chemically conjugated to the SO6 isoform of saporin. BU12-Saporin has demonstrable selective cytotoxicity against CD19^+^ leukaemia and lymphoma cell lines both in vitro and in vivo [50]. CD19 is found on the surfaces of all normal B-lymphocytes and the majority of B-cell malignancies but is not expressed by any other normal tissues, making it an ideal target for the majority of B-cell malignancies [51].

The safety and tolerability of BU12-SAPORIN was evaluated in eight adult B-cell lymphoma patients with advanced relapsed/refractory disease in a phase I dose escalation study [38]. Cohorts of three patients each received a dose of either 30, 60 or 120 μg/m^2^/day for seven days administered as a one-hour saline infusion. The study failed to establish the MTD only because of early closure due patient recruitment difficulties at the two participating centres. There were no definable complete responses (CRs) or partial responses (PRs), but one patient treated at the 120 mcg/m^2^ dose level showed a complete clearance of lymphoma cells from bone marrow. The only tangible toxicity observed was transient hepatotoxicity in patients treated at the 60 and 120 mcg/m^2^ dose levels that resolved spontaneously after cessation of treatment. There were no recorded cases of hypoalbuminemia or overt VLS. Despite the fact that BU12-SAPORIN is comprised of two unmodified non-human proteins (mouse IgG_1_ antibody and saporin), none of the treated patients developed HAMA or HASA responses to the IT, as measured over the duration of the study. This is surprising but can probably be accounted for by the fact that all were likely severely immunocompromised following heavy pre-treatment with prior cytotoxic therapy. Though this phase I study was terminated prematurely before a MTD was established, it did demonstrate that, for this limited size cohort, BU12-SAPORIN was safe to use up to a dose level of 120 μg/m^2^/day. The safety data that this limited phase I study provided subsequently paved the way for the approval of a phase I dose escalation study with BU12-SAPORIN in paediatric patients with CD19^+^ pre-B-cell ALL described in next section.

### 3.4. Phase I Study with BU12-SAPORIN in Relapsed Paediatric Pre-B-Cell Acute Lymphoblastic Leukaemia

On the basis of the safety and toxicity profile for BU12-SAPORIN acquired from the phase I study in adults with B-cell lymphomas described above, a separate phase I trial was designed and initiated in paediatric patients with relapsed or refractory pre-B-cell ALL under UKCCSG protocol NAG 2001-01 [40]. A total of five patients were recruited, all with advanced disease and all with a history of multiple relapses and/or chemotherapy refractory disease. In this dose escalation study, a starting dose of 240 mcg/m^2^/day for a five-day course of treatment was set and administered i.v. each day as an infusion in 100 mL of normal saline. Of the five patients entered onto the study, only two received the full dose over five days. The remaining three patients were withdrawn before receiving a full course of treatment because of complications due to a rapidly progressive disease or, in one case, to non-drug-related declining serum albumin levels. There were no drug-related adverse events or obvious toxicities and no evidence of VLS in any patient. There were no obvious disease responses, and all patients eventually succumbed to their disease, with survival times ranging from 10 days to 14 weeks following the cessation of BU12-SAPORIN treatment. The limited number of blood samples available posttreatment for each patient made an accurate evaluation of HAMA/HASA responses difficult, though suffice it to say, no immune responses against BU12-SAPORIN were detected in this small cohort of patients. The extremely friable and poor state of this patient population made the study extremely difficult to conduct and evaluate, and the investigators decided to close the trial prematurely and redesign the protocol to include a block of reinduction chemotherapy prior to receiving BU12-SAPORIN treatment. The rationale here was that having patients in CR or PR prior to receiving BU12-SAPORIN would help prevent patients from rapidly deteriorating due to disease progression, as experienced in the first study, thereby making drug safety evaluation more straightforward and accurate. However, following the redesign of the study, drug supply issues arose, because a new European Union Clinical Trial Directive required the use of a full cGMP compliant drug that was not readily available, and unfortunately, the redesigned study was never implemented.

### 3.5. Phase I Study of the Anti-CD38 IT OKT10-SAPORIN in Multiple Myeloma

OKT10-SAPORIN is a chemically constructed IT directed against the CD38 molecule found on a variety of normal lymphoid and other cell types. CD38 is highly expressed on normal and malignant plasma cells, making this a particularly attractive target in multiple myeloma (MM). Preclinical studies have shown that OKT10-SAPORIN is selectively cytotoxic for malignant B- and T-lymphocytes in vitro and in vivo in animal models [52,53]. A phase I dose escalation study [39] was conducted in a cohort of ten patients with CD38^+^ multiple myeloma. The primary purpose was to determine the safety, toxicity and pharmacokinetic profiles together with an MTD for OKT10-SAPORIN, with the secondary purpose of evaluating any disease response. Prior to the commencement of this formal phase I study under the UK’s former Doctor and Dentist Exemption (DDX) certificate study, four patients had already received OKT10-SAPORIN on an informally named patient basis, and this identified visual disturbances as being a potential side effect for this IT. Accordingly, special ophthalmological investigations were incorporated into the formal phase I study protocol [39] in order to evaluate patients prior to and after receiving OKT10-SAPORIN for any visual abnormalities. In this dose escalation study, cohorts of three MM patients with heavily pre-treated advanced disease received doses of 20, 30 and 40mcg/kg/day for a five-day course given as an i.v. infusion in 100 mL normal saline. A tenth patient also received OKT10-SAPORIN at the 40 mcg/kg/day dose level. Two patients treated at the 40 mcg/kg dose level withdrew from the study after three and four doses, respectively, because of blurred vision or raised liver function tests in the former case and after the fourth dose in the patient with blurred vision at their own request. The most common clinical adverse features were post-infusion pyrexia, fatigue with flu-like symptoms and nausea. Three patients experienced blurred vision during treatment. Biochemically, the most common adverse feature considered to very likely be due to the IT was hepatotoxicity (grades 1–3) experienced by all patients but which completely resolved following the cessation of treatment. Other less common toxicities included hypoalbuminemia, hyponatraemia, thrombocytopaenia, neutropenia and lymphopenia, all of which resolved spontaneously. There were no dose-limiting toxicities (DLTs), and the MTD was not established in this study. Only seven patients were evaluable for HAMA/HASA responses, and six (86%) of these seroconverted to become HAMA-positive. None of the seven made a detectable HASA response. Treatment with OKT10-SAPORIN at any of the dose levels gave no discernible improvement in the disease status for any of the patients. Unfortunately, the study was paused for almost a year over concerns about inappropriate tissue targeting, which was later shown to be unfounded but from which the study was not able to recover.

### 3.6. Phase I Study of a SubstanceP-Saporin TT in Pain Control for Cancer Patients

A novel approach for the use of a substanceP-saporin conjugate (SP-SAP) [54] to control the intractable pain in cancer patients was described by Frankel et al. in 2014 [36]. SP-SAP is a neuromodulator that binds to neurones, expressing NK-1 receptors normally found in the dorsal root of the spinal cord. Preclinical studies of SP-SAP in rats have demonstrated reductions in chemically or thermally induced pain, hyperalgesia and mechanical allodynia without any motor, sensory or behavioural dysfunction. Additionally, SP-SAP has been shown to reduce bone pain in dogs when injected intrathecally into the lumbar area without any attendant neurological, behavioural or histological changes [55].

The preliminary results of a phase I study with SP-SAP in cancer patients with intractable pain were reported in abstract form by Frankel et al. [36], as conducted under clinical trial reference number NCT02036281. This was a single subject cohort dose escalation study using doubling doses (1–64 mcg and a final dose of 90 mcg) of SP-SAP injected intrathecally into the L5-S1 interspace. Although not stated directly in the abstract, it can be inferred that only three patients were treated up to a dose level of 4 mcg. There are no results posted for NCT02036281 on the clinicaltrials.gov website, but the authors reported that no toxicities, neurological or cardiac abnormalities were observed and that, at the dose levels so far studied, there was no evidence of pain reduction in any of the treated patients. It was reported that the dose escalation study is continuing, but we were not able to locate any further information.

## 4. Pokeweed Antiviral Protein (PAP)-Based TTs

### 4.1. B43-PAP

B43-PAP is a second-generation IT comprised of a whole IgG_1_ murine human anti-CD19 monoclonal antibody (B43) chemically conjugated via a disulphide bond to the plant RIP pokeweed antiviral protein (PAP) [56]. This IT has been extensively studied in preclinical studies and demonstrates selective cytotoxicity for CD19^+^ malignant cell lines and primary B-cell malignancies in vitro [57] and in vivo [58,59]. The published clinical experiences with B43-PAP have been limited exclusively to paediatric pre-B-cell ALL.

### 4.2. B43-PAP in Paediatric Acute Lymphoblastic Leukaemia

The first description of B43-PAP used in a phase I dose escalation study in CD19^+^ pre-B-cell ALL was reported briefly in a review paper by Uckun [60]. Here, 17 paediatric pre-B ALL patients were treated at 11 different dose levels (range 0.5–1250 μg/kg body wt) via a 1-h i.v. infusion. It is not known if B43-PAP was given as monotherapy or given alongside other cytotoxic therapy in this study, as this was not stated. The MTD was not reached, with VLS and myalgias recorded as observed toxicities but with no hepatic, renal, cardiac or neurological impairments. None of the treated patients developed HAMA or HATA. The author reported that four CRs and one PR were seen, though the report did not specify at which dose levels. Five additional patients were reported to show demonstrable reductions in circulating peripheral leukemic blasts, but again, the dose levels and magnitude of the reductions were not reported in full and therefore remain unknown.

The next clinical uses of B43-PAP were reported in 1999 and 2000, respectively, as two separate meeting abstracts that described its use in combination with the standard four-drug reinduction treatment (VPLD) in paediatric cases of relapsed [61] or newly diagnosed pre-B-cell ALL [62]. The earlier 1999 phase I study reported by Seibel et al. [61] explored the safety and activity of B43-PAP at two different dose levels (1 and 1.5 mg/m^2^/day given daily on days 9–13 and 21–25) given concomitantly during reinduction with standard VPLD chemotherapy to 24 children with relapsed CD19^+^ ALL. It was reported that 18 were evaluable for toxicity and 15 for response. One patient treated at a dose level of 1 mg/m^2^/day experienced grade 4 myalgias, whilst DLT was seen in three out of eleven patients, two of which developed grade 3 or 4 hepatotoxicity. Ten CRs and two PRs were reported, whilst three patients had progressive disease or had no response. This data allowed the authors to conclude that B43-PAP could be safely administered in combination with VPLD reinduction chemotherapy to children with CD19^+^ ALL, and they announced that this would form the basis of the rationale to add B43-PAP to one of the treatment arms of a new children’s cancer group protocol for a cohort of newly diagnosed high risk paediatric cases of CD19^+^ ALL. This subsequently led to an upfront study with B43-PAP used in combination with four-drug (VPLD) induction chemotherapy in newly diagnosed cases of paediatric pre-B-cell ALL who were slow early responders in order to determine whether the addition of this IT to the standard induction protocols improved the response rate in comparison with the children who did not receive it [62]. To achieve this goal, 28 patients with high-risk ALL were randomised to receive B43-PAP or not, together with standard VPLD induction chemotherapy. The IT was administered at a single dose level of 1 mg/m^2^/day for five consecutive days (days 9–13), commencing nine days after the start of VPLD induction treatment. In addition, a comparator group of 255 nonrandomised patients with similar disease features received VPLD induction therapy alone. Patients who received both VPLD + B43-PAP had significantly better reductions in bone marrow leukemic blasts than either the randomised or nonrandomised cohorts, and the authors concluded that this outcome merited further investigation in a larger number of patients.

Following this, Seibel and her co-workers published their results for an extended study with the B43-PAP IT in 30 paediatric patients with relapsed/refractory pre-B ALL (US Children’s Oncology Group protocol CCG-0957). There, B43-PAP was used in a 3 + 3 dose escalation design using just three dose levels (1, 1.5 and 2.0 mg/m^2^/day given i.v. daily on days 9–13 and 21–25) to paediatric patients with relapsed/refractory pre-B ALL. The IT was administered together with either a three-drug (VPL) or four-drug reinduction treatment given over 28 days. This study was published in full detail by Meany et al. [37] in 2015. Grade III/IV toxicities were encountered and included myalgias, motor dysfunction, pulmonary toxicity and elevated liver transaminases. Grade III/IV VLS was seen in eight patients, but only one reached grade IV severity, resulting in pulmonary DLT and multiorgan failure. The authors suggested that the relatively low incidence of severe VLS in their study was the result of aggressive pre-emptive fluid and medication interventions, as specified in the trial protocol. Dose-limiting toxicities occurred only in patients receiving the four-drug induction regime when given together with B43-PAP, and therefore, the MTD for the three-drug regime was not established in this study. Of the 20 patients that were evaluable, 14 achieved CR by 28 days. HAMA and HATA responses were not evaluated in this study. The authors concluded that B43-PAP could be administered safely together with standard induction chemotherapy and that the IT exhibited clinical anti-leukaemia activity in this setting. It is unfortunate that, despite the encouraging results that this study provided, it was never progressed to a further phase II or III study because of a B43-PAP drug supply issue. Consequently, the full therapeutic benefits of B43-PAP in paediatric CD19^+^ ALL have not been fully explored.

## 5. Gelonin-Based TTs

### Phase I Study with the IT HUM-195/rGEL in AML

HUM-195/rGEL is a second-generation IT constructed by site-specific chemical conjugation via a cleavable disulphide bond of the humanised anti-CD33 antibody HUM-195 to a recombinant bacterially expressed version of Gelonin (rGEL) [63,64]. Gelonin, whilst possessing equivalent RIP activity to other plant-derived RIPs such as saporin, appears to have lower intrinsic nonspecific in vivo toxicity [65]. HUM-195/rGEL has been shown to be highly effective at selectively killing a CD33^+^ acute myeloid leukaemia (AML) cell line both in vitro and in vivo in an animal model of AML [64,66]. CD33 is expressed on the majority of malignant myeloid cells [67] but is absent from the myeloid stem cell population [68,69]. This means, in principle, that myeloid cells killed by IT HUM-195/rGEL when administered to patients would be subsequently replenished from earlier myeloid precursors that lack CD33 expression.

To primarily test the safety and secondly response rates, Borthakur et al. [35] undertook a phase I dose escalation study with HUM-195/rGEL in 27 patients with advanced AML and 1 with myelodysplasia under clinical trial protocol NTC00038051.

Four dose levels: 12, 18, 28 and 40 mg/m^2^ per course of HUM-195/rGEL were investigated in a 3 + 3 study design [35]. Dose-limiting toxicities seen post-infusion were allergic-type reactions that included hypoxia and hypotension. The MTD was established as the 28 mg/m^2^ dose level. Four patients exhibited a 50% or greater reduction in peripheral blood blasts, while three patients treated at the 10, 12 and 28 mg/m^2^ dose levels showed a 38–50% reduction in bone marrow blasts. Detectable immune responses against HUM-195/rGEL were low, with only 2 out of 23 patients developing antibodies against the gelonin component. These were heavily pre-treated patients and were likely to be somewhat immunocompromised, making it difficult to evaluate how effective the deimmunisation of the gelonin component of the IT had actually been. Importantly, this study demonstrated that HUM-195/rGEL can safely be administered to AML patients with advanced disease with an established MTD of 28 mg/m^2^. Anti-leukaemia activity was demonstrable in 7 out of 27 (26%) patients, but unfortunately, phase II or III studies have not been undertaken due to subsequent drug supply issues.

## 6. Bouganin-Based TTs

VB6-845 is a fully recombinant IT consisting of a fusion between a humanised single-chain antibody fragment (scFv) with specificity for the epithelial cell adhesion molecule (EpCAM) and a deimmunised version of the RIP bouganin [21,70]. Deimmunisation of bouganin was achieved by identification and removal by the genetic engineering of putative T-cell epitopes without affecting the catalytic potency of the RIP [70]. A wide range of carcinomas, which, by definition, are of epithelial origin, express EpCAM on their surface, and this IT is designed to target such tumours. The results of a preliminary phase I dose escalation study were reported briefly by Kowalski et al. in 11 patients with a range of solid epithelial tumours [33]. Cohorts of three to six patients for each dose escalation were based on a modified Fibonacci design with a starting dose of 1 mg/kg body weight progressing to 2.0 mg/kg. Treatment was given once weekly in four-week cycles until unacceptable toxicity occurred, until all lesions disappeared or until the disease progressed. The MTD was not reached in this study, and the authors indicated that additional patients would need to be recruited in order to determine this. A single DLT of an infusion reaction was noted at a dose level of 2.0 mg/kg. Other toxicities observed were fever, hypotension and hypoalbuminemia. Immunogenicity data obtained for ten patients revealed very low anti-bouganin responses below or near the limit of detection without specifying the method employed.

A later publication in the form of a book chapter from the same group [34] briefly described a phase I dose escalation study with the VB6-845 IT in 15 patients with various EpCAM-positive solid tumours that included renal, ovary, breast, gastric, pancreas, non-small cell lung and colorectal cancers, primarily to determine the MTD and evaluate the safety and tolerability of the IT and secondarily to assess the immunogenicity, pharmacokinetic profile and any antitumour effects (clintrials.gov NCT 00481936). Cohorts of three patients, each received monotherapy-escalating doses of the drug based on a modified Fibonacci scheme (1, 2 or 3.34 mg/kg dose) of VB6-845 administered as an i.v. infusion once weekly in 4-weekly cycles. The maximum treatment duration was 16 weeks. There was only one DLT of a grade 4 acute infusion reaction reported as occurring in one patient from cohort 2 treated at 2.0 mg/kg. The MTD was not established as the final dose level of 3.34 mg/kg due to early termination of the study. There were a total of five serious adverse events (SAEs) reported, two of which were infusion-related and characterised by hypotension, fever and nausea. At least ten other patients experienced mild/moderate adverse events (AEs) that resolved within 1 to 2 days. Pyrexia was commonly reported. The preliminary efficacy data were encouraging, with five patients demonstrating stable disease and two a decrease in measurable tumour sizes in two patients from the second dose cohort (renal cell and breast carcinoma). The immunogenicity of VB6-845 in this patient population was reported to be very low, attesting to the potential value of bouganin deimmunisation as a strategy to overcome the immunogenicity problem. No further reports on the clinical use of VB6-845 have been found, and this IT is now listed in the portfolio of Sesen Bio (www.sesenbio.com (accessed on 15 June 2022)).

## 7. Momordin-Based TTs

BDI-1-MD is a first-generation IT comprised of the IgG_1_ murine monoclonal antibody BDI-1 with specificity for a bladder cancer-associated antigen [71] conjugated chemically to the glycosylated RIP momordin [20]. Yu et al. [32] very briefly described a series of 18 patients with superficial transitional cell carcinoma of the bladder treated intravesically with an unspecified dose of BDI-1-MD. Patients were followed up for between 8 and 24 months (mean 12.5 months) following the initial treatment. These workers reported that the IT was effective at clearing small tumours <1 cm in size in all but one of the 18 patients. Details of the study were, however, very sparse, and it is difficult to judge the precise outcomes that were achieved from the limited information given in their paper.

## 8. Overcoming Practical and Clinical Problems with TTs

TTs are large macromolecular structures and, as such, present special challenges when it comes to the manufacture, formulation, storage, administration and potential toxicity issues in the patient. These issues differ significantly from those encountered for the common small molecule cytotoxic drugs used in cancer therapy. This is particularly pertinent when comparing the non-overlapping pharmacological mechanism of action of small molecule cytotoxic drugs and enzymatic RIP activity possessed by TTs. This lack of overlap theoretically sidesteps any emerging acquired resistance to small molecule cytotoxic drugs, with multidrug-resistant tumour clones still remaining sensitive to TT-mediated killing. This forms one of the most important arguments in favour of the clinical use of TTs as an adjunct to conventional chemotherapy, where there is some compelling experimental evidence that they also may also act as chemo/immuno-sensitisers [72,73].

### Immunogenicity

TTs are comprised of at least two macromolecular protein-based structures and therefore display multiple epitopes that can potentially elicit an immune response in immunocompetent drug recipients. Unless the recipient patient has been exposed to the same or similar cross-reactive protein structures in the past, this is rarely a problem during the first administration; rather, it becomes a problem for subsequent administrations of the TT; following which, the recipient mounts secondary and tertiary immune responses. This leads to the production of anti-TT antibodies that may be directed against the target ligand-binding domain (e.g., antibody or growth factor), the toxin component or both and are commonly referred to as HAMA or human antitoxin antibody (HATA) responses, respectively. This can result in the neutralisation and/or rapid removal of the TT from the circulation and, in some instances, to potential immunopathological consequences for the patient from the deposition of immune complexes in life-sustaining tissues or via activation of other immunopathological mechanisms. Furthermore, and most importantly, HAMA and HATA antibodies may neutralise TT efficacy by binding to the ligand-binding site, thus preventing or reducing attachment to the target cell or by binding to the enzymatic active site of the toxin or both.

## 9. Factors Limiting the Clinical Use of TTs

### 9.1. Immunogenicity and Toxicities

Early generation ITs that utilised native mouse antibodies chemically coupled to native proteinaceous toxins were strongly immunogenic in humans being readily recognised as non-self by immunocompetent patients receiving these. This was a major limitation for the clinical use of early generation TTs in humans, as they elicited primary, secondary and tertiary immune responses in human patients receiving multiple doses of IT [74,75]. Recipient anti-TT antibody responses against either the ligand-binding domain or toxin domain can reduce or neutralise the TT therapeutic potency by either increasing the rate of clearance from the blood or by blocking the ligand binding or toxin catalytic sites via steric hindrance due to bound anti-TT antibodies produced by the recipient. Strategies towards reducing the immunogenicity of TTs have varied and ranged from the use of immunosuppressive drugs to the attempted sequestration of immunogenic epitopes displayed by the TT with agents such as polyethylene glycol (PEG) [11]. However, perhaps the most promising way forward is offered by deimmunisation of the toxin component by the identification and removal of T- and/or B-cell epitopes from the whole molecule using recombinant technology without affecting the toxin enzymatic activity. Proof that this approach works for bacterial toxins such as diphtheria [76] and pseudomonas [77] has already been demonstrated and also achieved for the plant-derived type I RIP bouganin [70]. In all cases, significant reductions in the immunogenicity and subsequent host HATA responses have been observed.

The ligand-binding domain component of a TT, if derived from a non-human source, faces the same issue of unwanted immunogenicity. In the case of the targeting domain being formed by a wholly human growth factor or other human protein or protein fragment, this is rarely a problem. The humanisation of murine antibodies and the development of new genetic techniques that allow the production of fully human antibodies have ameliorated this problem considerably [78]. However, even fully human antibodies can elicit an unwanted immune response against paratopes in the antibody hypervariable region, and therefore, a complete lack of immunogenicity cannot be guaranteed, even for fully human antibodies or their fragments [79].

### 9.2. Target Antigen Suitability

A major consideration for the success of any TT treatment is the suitability of the target molecule that is selected for delivery of the toxin payload. The expression level and heterogeneity of expression within the global tumour cell population, together with an internalisation characteristic that ensures delivery of the toxin component to an appropriate intracellular compartment within the target cell, all play a crucial role, ultimately determining whether the tumour is successfully ablated or not. It is also of critical importance that the target antigen selected is not expressed by any normal life-sustaining tissue to avoid potentially catastrophic toxicity problems. This was exemplified previously in women with breast cancer treated with a ricin A chain IT targeting an antigen on breast cancer cells that was cross-reactive with Schwann cells, resulting in debilitating neurological consequences [80]. Of particular importance is the selection of a target molecules that is expressed not just by the bulk of tumour cells but also by the tumour stem cell that continuously gives rise to a non-self-renewing cancer cell progeny. Failure to target and eliminate the tumour stem cell will almost inevitably result in tumour recurrence. The issue of target molecule heterogeneity across the entire bulk tumour cell population is also an important issue, as tumour cells that are downregulated or lack the expression of any single target molecule (whether due to negative selection under pressure or other epigenetically driven factors) will evade killing by a TT that targets only that single molecule. One possible solution to this problem is to target two, three or more antigens simultaneously on the tumour cell surface. This has the effect of increasing the statistical probability of reaching all the cells within the tumour based on the premise that cells negative for one target molecule will likely be positive for others [81]. Theoretically, the greater the number of target molecules aimed at simultaneously, the less likely it becomes for “escapee” tumour cells to emerge. There is good experimental evidence from *in vivo* animal model studies that combination targeting is indeed significantly better than single antigen targeting [53,82,83,84], though the three separate phase I clinical studies with Combotox, a ricin A chain-based IT cocktail simultaneously targeting CD19 and CD22, were inconclusive, as the early phase studies were designed to primarily evaluate safety rather than response [10,85,86]. Further exploration of the clinical value of the combination targeting approach with TTs in cancer (and, indeed, other antibody-based therapies) are urgently required if the potential of this approach is to be fully realised.

### 9.3. Manufacturing and Formulation Considerations

As we alluded to earlier in this review, TTs produced as recombinant fusion proteins have some major advantages over chemical constructs. Firstly, they are structurally more consistent and therefore more homogeneous in the final product, making an assessment of their toxicity profile and therapeutic activity much more reliable than a final product that has a heterogeneous molecular content. Secondly, manufacturing a TT from genetically encoded material using fermentation technology lends itself to facile high-capacity production with the minimum steps required to achieve a final product in contrast to the multistep processes required for ADCs and other similar chemically constructed conjugates.

Prokaryotic bacterial or eukaryotic systems have both been used for the expression of recombinant TTs. For eukaryotic expression systems, there have been past concerns that the toxin domain may compromise the expression efficiency by auto-poisoning the expression vector’s ribosomes, thus limiting the level of achievable translation and, hence, protein production. Careful optimisation of such systems has fortunately largely overcome this concern. Saporin [87], gelonin [63] and bouganin [88] have all been successfully expressed in *E.coli* together with an FGF-Saporin recombinant TT [89]. Saporin [1] and PAP [90] have both been expressed in the methylotrophic yeast *Pichia pastoris* following codon optimisation for this expression vector [91]. If high yield efficiencies can be reliably achieved in a eukaryotic expression system, such as *P. pastoris*, then such systems would be infinitely more preferable to any prokaryotic expression system. This is because misfolded recombinant proteins accumulating in the bacterial periplasmic space necessitates additional labour-intensive procedures to extract and refold the misfolded protein in order to obtain biologically active TTs. In contrast, recombinant proteins, including TTs expressed in *P. pastoris*, are secreted correctly and folded directly into the surrounding fermentation medium thanks to the eukaryotic translational machinery that recapitulates the majority of the co- and posttranslational events necessary for the formation of secretion-competent polypeptides. Della Cristina et al. [3] successfully constructed a range of recombinant ITs comprised of an engineered anti-CD22 scFv fragment of the monoclonal antibody 4KB128 [92] fused to either full-length saporin or the PE40-truncated form of *Pseudomonas* exotoxin and successfully expressed these in *P. pastoris* following codon and other optimisations of the small-scale fermentation system used. Both the saporin- and PE40-based recombinant ITs obtained showed similar immunospecific cytotoxicity for the CD22^+^ human lymphoma cell line Daudi. This success was also similarly achieved for the TT ATF-Saporin by Provenzano et al. [93] for a fusion protein comprised of the N-terminal fragment of urokinase-type plasminogen activator (uPA) and saporin that targets the uPA receptor (uPAR) expressed at high levels on a variety of different cancer cells [94]. These authors constructed *P. pastoris* clones that expressed and secreted ATF-Saporin in a range from 1 to 6.99 mg/litres of fermentation medium, with the majority being in the 3.0–3.99 mg/litre range. Selective cytotoxicity of the obtained ATF-Saporin TTs was demonstrated on the uPAR-expressing cell line U937 [93].

The rationale in favour of using eukaryotic expression systems to manufacture TTs for clinical use is compelling, and the work briefly described above strongly supports the feasibility of this approach. Optimisations above and beyond those currently achieved in the limited studies just described will be necessary for the scaling up of fermentation processes, but we consider that this will be readily achievable if the appropriate resources and know-how are available.

TTs are large proteinaceous macromolecules that are prone to denaturing or micro-aggregation if handled incorrectly. Formulations to avoid such problems from the manufacturer to pharmacy to bedside will be needed to avoid problems that are an intrinsic property of any class of any macromolecular biologic and to avoid serious toxicological consequences in recipient patients, as have been described previously [10].

### 9.4. Strategies to Improve Therapeutic Index

Whilst it is beyond the scope of this review to undertake a detailed discussion of the methods that have been or may be used to improve the therapeutic index of TTs, it is worthwhile to touch on this as a subject area here, because it could well transform the clinical efficacy of TT therapies. There have been many different approaches to increasing the potency of TT cytotoxicity, ranging from small molecules that modify TT internalisation and intracellular processing to methods that mimic viral entry [95] or disrupt the endolysosomal system, facilitating the release of the TT into the target cell cytosol [96]. For useful reviews of the various methods that have been employed over the years to enhance the TT potency, the reader is referred to the reviews of Giasanti et al. [97] and Gilabert-Orial et al. [98].

## 10. Conclusions and Future Perspectives

TTs have been largely neglected in recent years in favour of ADCs and other immunotherapeutic or genetic therapeutic interventions. This situation is grounded in the biases that have emerged over a four-decade period and have subsequently plagued TT development during this time. Arguably, one of the greatest strengths of TTs over ADCs is the fact that they are amenable to facile design and manufacture utilising recombinant technology, whereas ADCs generally are not. The clinical trials that have been described in this review attest to the fact that ITs can and do exhibit substantial clinical activity against the malignancies for which they were designed, and it is our opinion that, to date, their therapeutic potential has not been sufficiently or adequately investigated. This has been due largely to resource issues rather than to any serious drug-related problems. One major obstacle that has proven a major impediment for the majority of studies described in this review has been concerned with drug supply issues and the lack of availability of drugs (for varied reasons, ranging from cost issues to new regulatory requirements) to complete an ongoing phase I trial or to progress to later phase studies. This has had the effect of halting the early phase clinical studies that initially demonstrated the early evidence of antitumour activity before they were fully concluded. This prevented the progress to later phase studies that would have determined their efficacy. These unfortunate experiences make it obviously clear that a robust and reliable pipeline for the drug supply is absolutely essential if future clinical studies are to succeed. This will likely only be achieved when a firm commitment is made by commercially driven companies willing to take on the risk, as is the case for any new drug under development.

Previous concerns have been raised about the transient nature of responses that have been achieved with ITs, and this has led to a perception that this class of drug is therefore of limited value clinically. Then, we might also conclude so too are the majority of mainstream anticancer drugs, which, when used as a monotherapy, also fail to achieve deep and lasting responses but then do so when used in combination with other cytotoxic drugs that have non-overlapping mechanisms of action. This is also likely to be true for TTs when used in combination with other small molecule drugs and biologics for which ample supportive preclinical experimental evidence already exists [73,99,100]. Furthermore, TT-based treatments are likely to be at their most effective when used as an adjunct therapy alongside other therapeutic modalities, such as surgical/radiotherapy debulking and immuno/chemotherapy as a means to eliminate microscopic levels of disease. With further expeditious and suitably resourced investigation, TTs could still be set to emerge as an important mainstream therapeutic for cancer in the years that lie ahead.

## Figures and Tables

**Table 1 toxins-14-00563-t001:** A list of clinical studies undertaken with TTs incorporating plant-derived single-chain ribosome inactivating proteins.

Study Type	Disease	No.	TT Name(Type)	Molecular Target(s)	RIP	Comment	Ref.
Pilot	Hodgkin’s Lymphoma	4	Ber-H2/SO6 (immunotoxin) ^+^	CD30	Saporin (SO6 isoform)	Major Responses observed	Falini et al. (1992) [29]
Pilot	B-cell Lymphoma & B-cell CLL	2	Un-named (BsAb) *	CD19 or CD22 × Saporin	Saporin (SO6 isoform)	Response observed with anti-CD22 BsAb	Bonardi et al. (1992) [30]
Pilot	B-cell lymphoma & CLL	5	Un-named (mixture of 2 BsAbs)	CD22 × Anti-Saporin	Saporin (SO6 isoform)	Responses observed	French et al. (1996) [31]
Pilot	Bladder Cancer	18	BDI-1-MD^+^	Bladder Ca Antigen	Momordin	Responses observed	Yu et al. (1990) [32]
Phase I	Carcinomas	15	VB6-845 (deimmunised recombinant immunotoxin) ^+^	EpCAM	Bouganin	Responses observed	Kowalski et al. (2008) [33] Entwistle et al. (2013) [34]
Phase I (dose escalation)	AML	28	HUM-195/rGEL (cc immunotoxin) ^+^	CD33	Gelonin (recombinant)	Responses observed	Borthakur et al. (2012) [35]
Phase I	Cancer (Pain control)	23	SP-SAP (Targeted Toxin)	Substance *P*	Saponin	No pain control observed	Frankel et al. (2013) [36]
Phase I/II (dose escalation)	B-cell ALL	30	B43-PAP (cc immunotoxin) ^+^	CD19	Pokeweed Antiviral Protein (PAP)	Major responses	Meany et al. (2015) [37]
Phase I (dose escalation)	B-cell Lymphoma	8	BU12-SAPORIN (cc immunotoxin) ^+^	CD19	Saporin (SO6 isoform)	Minor responses	Sweetenham & Flavell (1999) (unpublished) [38]
Phase I (dose escalation)	Multiple Myeloma	10	OKT10-SAPORIN (cc immunotoxin) ^+^	CD38	Saporin (SO6 isoform)	Minor responses	Samson et al. (2000) (unpublished) [39]
Phase I (dose escalation)	Paediatric B-cell ALL	5	BU12-SAPORIN (cc immunotoxin) ^+^	CD19	Saporin (SO6 isoform)	Minor responses	Morland & Flavell (2002) (unpublished) [40]

* BsAb, Bispecific antibody; ^+^ cc immunotoxin, chemically constructed immunotoxin.

## Data Availability

Not applicable.

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
