# Peer review of "Plant-Derived Type I Ribosome Inactivating Protein-Based Targeted Toxins: A Review of the Clinical Experience"

_toxins, 2022, doi:10.3390/toxins14080563_

Round 1

Reviewer 1 Report

I found the review of plant derived type I Ribosome inactivating Protein-based targeted toxins informative and very well written. I agree that despite the fact that there has been significant progress in this area, over the past several years these agents have been neglected and perhaps it is time to take a closer look at reinventing them utilizing our current scientific progress.  I think the readership of Toxins will find this review interesting. In fact, I believe it will be appealing to the antibody targeting readership at large. It does fill an important niche since I myself knew of these very interesting catalytic targeted toxins and had wondered about the final clinical results.  Many preclinical papers were published with no clinical followup. 

My comments are rather minor. 

-There is an extra period on line 9. 

-My understanding is that many of these RIP based targeted toxins have an identical catalytic mechanism of action and that is recognizing a universally conserved stem-loop structure in 23S/25S/28S rRNA, depurinating a single adenine (A4324 in rat) and irreversibly blocking protein translation, leading finally to cell death of intoxicated mammalian cells. It would be helpful to explain that they are enzymes early on and perhaps include a Figure illustrating this mechanism. In fact, it would be helpful to include a figure diagram since the manuscript has no Figures at all.  

-Next, the authors rightfully state that the many of the trials were discontinued based on “supply” issues. In the final paragraphs where the authors discuss “supply” as an issue, I think they could include a discussion of factors that that they believe might influence supply. 

-Instead of reference number 43, the first report of B43-PAP in ALL was published in 1986 in the high profile “journal of experimental medicine”. 1986 Feb 1;163(2):347-68. This reference should be used instead. 

The organization and length of the article is fine. I think the manuscript should be published if some of these minor issues are addressed.I found the review of plant derived type I Ribosome inactivating Protein-based targeted toxins informative and very well written. I agree that despite the fact that there has been significant progress in this area, over the past several years these agents have been neglected and perhaps it is time to take a closer look at reinventing them utilizing our current scientific progress.  I think the readership of Toxins will find this review interesting. In fact, I believe it will be appealing to the antibody targeting readership at large. It does fill an important niche since I myself knew of these very interesting catalytic targeted toxins and had wondered about the final clinical results.  Many preclinical papers were published with no clinical followup. 

My comments are rather minor. 

-There is an extra period on page 9. 

-My understanding is that many of these RIP based targeted toxins have an identical catalytic mechanism of action and that is recognizing a universally conserved stem-loop structure in 23S/25S/28S rRNA, depurinating a single adenine (A4324 in rat) and irreversibly blocking protein translation, leading finally to cell death of intoxicated mammalian cells. It would be helpful to explain that they are enzymes early on and perhaps include a Figure illustrating this mechanism. In fact, it would be helpful to include a figure diagram since the manuscript has no Figures at all.  

-Next, the authors rightfully state that the many of the trials were discontinued based on “supply” issues. In the final paragraphs where the authors discuss “supply” as an issue, I think they could include a discussion of factors that that they believe might influence supply. 

-Instead of reference number 43, the first report of B43-PAP in ALL was published in 1986 in the high profile “journal of experimental medicine”. 1986 Feb 1;163(2):347-68. This reference should be used instead. 

The organization and length of the article is fine. I think the manuscript should be published if some of these minor issues are addressed.

Author Response

We thank this reviewer for their helpful and positive comments to which we respond as follows:-

The mechanism of action of RIP's has already been briefly described early in the text under section 2 which we have now also expanded on a little more. The purpose of this review is to highlight clinical studies and not to do any in depth treatise on mechanisms of action and we feel that it is sufficient that the paper refers to other studies that deal with this subject in greater detail. For the same reason we do not feel that adding a diagram of mechanism(s) of action adds anything to this review whose main purpose is to describe clinical studies and the practical problems that have been encountered.

We have added additional sentences to highlight the issues surrounding the drug supply problem.

We have replaced ref 43 (now ref 52) with Uckun et al 1986.

Reviewer 2 Report

This review gives an interesting historical and scientific synthesis of immunotoxin (IT) clinical applications based on type 1 RIPs. The article is well written and treats the actual problems limiting the clinical use of RIP based ITs and the advantages/strategies to overcome them. Most of the reported clinical studies stopped at phase 1 though the promising therapeutic benefits, and have not been further explored often due to drug supply issues or to limited number of patients. I think this review may contribute to highlight the substantial clinical efficacy of RIP based IT therapy and aid their further development. However, some revisions are necessary and some points should be clarified in order to improve the review.

Chapter “3.1. BER-H2-SO6 Immunotoxin (Anti-CD30)”. This chapter is very short and does not give information on the different steps of the research that led to in vivo patients’ treatment.

Namely:

-        Selection of saporin

Bolognesi, A., Tazzari, P.L., Tassi, C., Gromo, G., Gobbi, M. and Stirpe, F. - A comparison of anti-lymphocyte immunotoxins containing different ribosome-inactivating proteins and antibodies. Clin. Exp. Immunol. (1992) 89: 341-346.

-        In vitro results with BerH2/saporin conjugate

Tazzari, P.L., Bolognesi, A., De Totero, D., Falini, B., Lemoli, R.M., Soria, M.R., Pileri, S., Gobbi, M., Stein, H., Flenghi, L., Martelli, M.F., and Stirpe, F. - Ber-H2 (anti-CD30)- saporin immunotoxin: a new tool for the treatment of Hodgkin's disease and CD30+ lymphoma. In vitro evaluation. British J. Haematol. (1992) 81: 203-211.

-        Experimental therapy in four patients with refractory Hodgkin’s disease

Falini, B., Bolognesi, A., Flenghi, L., Tazzari, P.L., Broe, M.K., Stein., H., Dürkop, H., Aversa, F., Corneli, P., Pizzolo, G., Barbabietola, G., Sabattini, E., Pileri, S., Martelli, M.F. and Stirpe, F. - Response of refractory Hodgkin's disease to monoclonal anti-CD30 immunotoxin. Lancet (1992) 339: 1195- 1197.

-        Final data on 12 patients

Pasqualucci, L., Flenghi, L., Terenzi, A., Bolognesi, A., Stirpe F., Falini, B. - Immunotoxin therapy of haematopoietic malignancies. Haematologica  (1995) 80: 546 556)

Finally, the authors should add the information that immunotoxins was obtained by chemical conjugation (line 115); that the patients had advanced refractory Hodgkin’s disease (chemo- and radio-resistant) and information about the percentage of tumor reduction obtained in patients after treatment.

Line 91. The authors write “This has the effect of irreversibly inhibiting protein synthesis by preventing an association between elongation factors and the ribosomal machinery. This event leads to apoptosis of the intoxicated cell via induction of the ribotoxic stress response[26]”. The ribotoxic stress response is not the only mechanism induced by RIP intoxication. Many questions have remained opened about the mechanism of RIP-induced apoptosis and the relationship between RIP N-glycosylase activity and the apoptotic events.

The authors should better explain that:

Ribosome is not the only substrate of RIP action.

Apoptosis can be induced by Unfolded Protein Response; genomic DNA damage; oxidative stress.

Finally, the authors should explain that apoptosis is not the only death pathway triggered, but that necroptosis and autophagy are also involved.

Chapter “Saporin-Based TTs”. In some studies (references 36 and 39) there are no correspondence between the number of patients reported in the text and that summarized in the table 1. Please check.

Chapter “Gelonin-Based TTs”. In the study referred to [58] the number of patients is 28 and not 27. In table 1 the correct number is reported. Please check.

Table 1. Two clinical studies are reported for Bouganin (reference 60 and 61). For each study miss the number of patients (11 and 15, respectively).

Table 1. The clinical study reported for Momordin (Ma et al [62]) is not correct. The study reported referred to Yu at al [63].

Table 1. I suggest the author to reorganize the table by placing first all the studies on saporin in date order and then the other RIPs following the order of the text.

Line 41. Please report “Ricinus communis” in italics.

Line 81. Please write the genus of the plant in capital letter.

Author Response

We thank this reviewer for their helpful suggestions. We have extended the scope of 3.1 BER-H2-SAP section and included additional references to give a better historical perspective on the development and clinical results obtained for this immunotoxin also making it clearer that this is a chemically constructed IT.

It is not the purpose of this review to give an in depth account on the potential mechanisms of cell killing by RIPs, an area that remains somewhat controversial and beyond the scope of this review. We have however indicated briefly in the text other potential mechanisms of cell killing with appropriate references.

The lack of correspondence between the two references in the  text and table has been corrected.

The reviewer points out that the gelonin based IT quoted in ref 58 (now ref 60 since addition of other references to the paper) should be 28 and not 27 as quoted in the text. In actual fact the text states that there were 27 AML patients and 1 MDS patient so is correct.

The number of patients (15) for the VB6-845 Bouganin TT has now been inserted into the table.

Incorrect reference quoted for BDI--1-MD momordin has now been corrected (Yu et al [29]

We would prefer to maintain the tabulated trials in date order.

Ricinus communis now in italics

Genus now capitalised. 

Reviewer 3 Report

The authors reviewed the clinical trials using Targetted Toxins based on type I RIP. The authors proceed to an in depth analysis of all published and unpublished trials and try to find an explanation for why these strategies have low success though their recognized potential. The studies reviewed are clearly presented and the manuscript is well written.

However, paragraphs 8 (there is only a 8.1 subparagraph) and 9-1, 9-2, 9-3 could be inserted after paragraph 2: it may thus help the reader for the critical evaluation of the clinical trials, in particular for problems concerning immunogenicity and production of TT. These points lead to serious limitations of the critical trials and could be introduced before.

Though the review is about type I RIP, the authors cite histotical approaches using ricinin line 40. However reference 5 does not belong to pioneering work using ricin, RTA or dgTRA. A more appropriate citation could be chosen (as PMID 1878584 or any other reference from the 90s)

Author Response

We thank this reviewer for their positive comments.

We would prefer to keep the order of sections 8 and 9 as they are, focussing the first part of the paper on the clinical trials themselves which then leads onto the other various issues. we do not believe that this will detract from an understanding of the problems associated with the clinical studies.

We have changed the reference referring to ricin work  to LeMaistre [ref 5] as requested. 

Reviewer 4 Report

The manuscript entitled “Plant-Derived Type I Ribosome Inactivating Protein-Based Targeted Toxins: A Review of the Clinical Experience” provided a nice review about the use of type I RIPs as targeted toxins for cancer disease. After a really brief introduction section, authors detail in a table, some studies performed with this system. In points 3 to 7, authors describe these studies according to the RIP, i.e. saporin, PAP, gelonin, bouganin and momordin. In point 8 and 9, authors discuss some of the main problems of these systems, such us, immunogenicity or unspecific toxicities. In the last point, authors discuss future perspectives in this frame.

Although the manuscript is well written, in my opinion authors should look for modern studies and compare Type I RIP TT systems with Type II A-Chain RIP TT systems, much more used in clinics.

Author Response

We thank this reviewer for their suggestion that our review should compare TT's constructed with type I and type II RIPs. As stated early in the text, the purpose of this review, is intended only to describe the clinical experience for type I RIP's and this suggestion therefore falls outside the intended scope of the review.

Round 2

Reviewer 2 Report

Please correct the two following references:

21. Bortolotti, M.; Bolognesi, A.; Polito, L., Bouganin, an Attractive Weapon for Immunotoxins. Toxins (Basel) 2018, 10, (8), 323

32. Tazzari, P. L.; Bolegnesi Bolognesi, A.; Totero, D. D.; Falini, B.; Lemoli, R. M.; Soria, M. R.; Pileri, S.; Gobbi, M.; Stein, H.; Flenghi, L.; Martelli, M. F.; Stirpe, F., Ber-H2 (anti-CD30)-saporin immunotoxin: a new tool for the treatment of Hodgkin's disease and CD30+ lymphoma: in vitro evaluation. British Journal of Haematology 1992, 81, 203-211.

Reviewer 4 Report

The version 1 of the manuscript entitled “Plant-Derived Type I Ribosome Inactivating Protein-Based Targeted Toxins: A Review of the Clinical Experience” included most of the suggestions of the different reviewers, not all of them. In your response you highlight that the objective of this review is "to describe the clinical experience for type I RIP's and this suggestion therefore falls outside the intend". Nevertheless, reviewer 3 and I suggested you that you also should include Type II A-Chain RIP TT systems.

Another important point is that you should look for modern studies to support your review. I only appreciate a little improvement in this aspect.